# Ternary Nanostructure Coupling Flip-Flap Origami-Based Aptasensor for the Detection of Dengue Virus Antigens

**DOI:** 10.3390/s24030801

**Published:** 2024-01-25

**Authors:** Mohd. Rahil Hasan, Saumitra Singh, Pradakshina Sharma, Chhaya Rawat, Manika Khanuja, Roberto Pilloton, Jagriti Narang

**Affiliations:** 1Department of Biotechnology, Jamia Hamdard, New Delhi 110062, India; rahilhasan789@gmail.com (M.R.H.); saumitrasingh1999@gmail.com (S.S.); pradakshinasharma@gmail.com (P.S.); chayarawat1553@gmail.com (C.R.); 2Centre of Nanoscience and Nanotechnology, Jamia Millia Islamia, New Delhi 110025, India; mkhanuja@jmi.ac.in; 3Institute of Crystallography, National Research Council, 00143 Rome, Italy

**Keywords:** origami, gold-decorated zinc and graphene nanocomposite, polyvalent dengue antigen, aptasensor, human serum

## Abstract

There is currently a lot of interest in the construction of point-of-care devices stemming from paper-based origami biosensors. These devices demonstrate how paper’s foldability permits the construction of sensitive, selective, user-friendly, intelligent, and maintainable analytical devices for the detection of several ailments. Herein, the first example of the electrochemical aptasensor-based polyvalent dengue viral antigen detection using the origami paper-folding method is presented. Coupling it with an aptamer leads to the development of a new notation known as OBAs, or origami-based aptasensor, that presents a multitude of advantages to the developed platform, such as assisting in safeguarding the sample from air-dust particles, providing confidentiality, and providing a closed chamber to the electrodes. In this paper, gold-decorated nanocomposites of zinc and graphene oxide (Au/ZnO/GO) were synthesized via the chemical method, and characterization was conducted by Scanning Electron Microscope, Transmission Electron Microscope, UV-Vis, and XRD which reveals the successful formation of nanocomposites, mainly helping to enhance the signal and specificity of the sensor by employing aptamers, since isolation and purification procedures are not required. The biosensor that is being demonstrated here is affordable, simple, and efficient. The reported biosensor is an OBA detection of polyvalent antigens of the dengue virus in human serum, presenting a good range from 0.0001 to 0.1 mg/mL with a limit of detection of 0.0001 mg/mL. The reported single-folding ori-aptasensor demonstrates exceptional sensitivity, specificity, and performance in human serum assays, and can also be used for the POC testing of various viral infections in remote areas and underdeveloped countries, as well as being potentially effective during outbreaks. Highlights: (1) First report on origami-based aptasensors for the detection of polyvalent antigens of DENV; (2) In-house construction of low-cost origami-based setup; (3) Gold-decorated zinc/graphene nanocomposite characterization was confirmed via FESEM/UV-Vis/FTIR; (4) Cross-reactivity of dengue-aptamer has been deduced; (5) Electrochemical validation was conducted through CV.

## 1. Introduction

The potential uses of paper substrates can be expanded by incorporating electronic applications into paper. Utilizing paper as a substrate for electronics can result in important advancements for papertronics applications. These applications have been used in diagnostic fields for the construction of electrochemical biosensors. Few reports are available on paper-based electrochemical biosensors in which paper, as a substrate, is utilized for the construction of electrodes, proving to be the appropriate substrate for low-cost printed microelectronics used with high-end silicon-based electronics [1]. Standard paper’s substantial surface roughness, porosity, and opacity all provide inherent challenges to electrical equipment placed on its surface [2]. These electrodes can be modified by employing a technique known as origami. The term “origami” has typically been linked solely to the art of paper folding and is based on the idea of folding flat sheets into intricate designs [3]. To the best of our knowledge, there is no report available on origami-based aptasensors (OBAs) for the detection of polyvalent antigens of the dengue virus. The dengue virus belongs to the Flaviviridae family. The Aedes aegypti mosquito transmits the dengue virus, which occurs as four antigenically distinct serotypes (DENV 1-4) and affects humans. Despite so much progress in the medical field, no effective treatment is available for the dengue virus. Proper diagnosis becomes the only hope for this. Many conventional diagnosis methods are available, such as ELISA, tissue culturing, immunofluorescence, and PCR. However, it has some drawbacks, including the fact that these tests are very costly, time-consuming, less precise, and demand specialists. A novel technology called a biosensor was developed to circumvent these constraints. It is inexpensive, fast, highly specific, and no expertise is required [4,5,6]. Electrochemical biosensors also come in various ranges based on their recognition elements [7,8,9], such as genosensors [10] and immunosensors [11]. Among these, the aptasensor is one of the promising and inexpensive biosensors. In aptasensor, the aptamer is used as a recognition element. An aptamer is a small single-stranded oligonucleotide sequence of DNA/RNA, and it can easily bind to the targets with great specificity and affinity. Aptamers are artificially devolved by many methods, such as SELEX, and using such affordable recognition elements (aptamer) leads to the construction of an effective diagnosis platform [12]. Along with biological recognition elements, nanomaterial is also playing a major role in the development of diagnosis platforms. Nanomaterials can be prepared in different forms such as composites of various nanomaterials, single-component nanomaterials, and hybrids. Composite nanomaterials show less toxicity, high signal amplification, high sensitivity, great electrical and thermal conductivity, good resolution assessments, improved solubility, and easier functionalization when compared to single nanomaterials [13]. Gold-decorated zinc and graphene oxide nanocomposites have been used for this experiment as they help in enhancing the signal, and are used to modify the working electrodes of the developed biosensor onto which aptamer was surface-immobilized, as this helps in increasing the functionality of the biosensor. This nanocomposite/aptamer-modified biosensor was used as a signal amplification platform for DENV diagnosis. The interaction of aptamer and DENV was discovered by lowering the current induced by the interaction of an anionic mediator, namely methylene blue (MB). The analytical response of the biosensor was evaluated using CV and then confirmed with a potentiostat.

In this work, the analytical performance of an origami-based aptasensor utilizing gold-decorated zinc and graphene oxide nanocomposites for the specific determination of polyvalent dengue antigens has been presented. To overcome the drawbacks of nanoparticles, we have incorporated a composite of three nanomaterials for the current research. Zinc has been used to increase biocompatibility and increase surface immobilization, and graphene has been utilized to enhance the conductivity of the sensor. Zinc-graphene nanomaterial is decorated with noble nanomaterials (AuNPs), as they are small in size, and thus imbibe onto the porous paper. Since the claimed functioning of the three-electrode system is more promising than that of the two-electrode system, as reported by Hasan et al. [14], we utilized a three-electrode system. The constructed sensor benefits significantly from the use of the origami platform, which increases the electrode’s design confidentiality, among other advantages. In commercial SPEs, an insulator layer is present, which is used for the safety purpose of the electrodes; therefore, the presented origami platform also contains a flap portion, which is used as an insulator layer that shields the electrode tracks from the external environment. All of these factors enhance the sensor’s capability of identifying as a POC device, making the suggested aptasensor an appealing option for precise and sensitive DENV detection.

## 2. Material and Methodology

### 2.1. Chemicals, Reagents, Apparatus

Conductive ink/paste (black and silver) were acquired from Snab Graphix Private Limited (Bengaluru, India) in order to be used in the fabrication of the origami design. MB was supplied by Sigma-Aldrich, (Bangalore, India), and all other chemicals were of AR quality. For the preparation of PBS, NaCl, potassium chloride, Na_2_HPO_4_, and KH_2_PO_4_ were purchased from LOBA (Wadesboro, NC, USA). To make solutions for the aptamer and antigen (pH-7.4, 100 M), PBS must be prepared.

The DENV-Apt, a 34-base oligonucleotide, was purchased from MTOR life science Pvt. Ltd. (Delhi, India) and was complementary to the polyvalent DENV antigen (ProspecTany TechnoGene Ltd., Rehovot, Israel).

Sequence of aptamer:

GCACCGGGCAGGACGTCCGGGGTCCTCGGGGGGC [15]

For the preparation of silver nanoparticles, silver nitrate (Thermo Fisher Scientific India Pvt. Ltd., Mumbai, India) and sodium borohydride (GLR Innovations, New Delhi, India) were utilized.

For the preparation of AuNPs, chloroauric acid (GLR Innovations), ascorbic acid (Sisco Research Laboratories Pvt. Ltd., Mumbai, India), tri–sodium citrate (Loba Chemie Pvt. Ltd., Mumbai, India), and sodium borohydride (GLR Innovations) were used.

The local hospital provided the human serum. To conduct electrochemical measurements, such as cyclic voltammetry, Metrohm Dropsens potentiostat (Stat-I 400s) (Dropsens, Oviedo, Spain) was utilized. Quanta 3D FEG (Philips, Cambridge, MA, USA) was used to study the material’s surface structure utilizing field exhaust scanning electron microscopy (FESEM) technology (FEI). The Agilent Cary100 series UV-Vis spectrometer (Agilent, Santa Clara, CA, USA) that was used to measure UV-Vis absorbance and FTIR (Optic Vertex spectrometer, FTIR, Zurich, Switzerland) were employed.

### 2.2. Pre-Treatment of Human Serum

Healthy subjects’ blood samples were kept for twenty minutes before being centrifuged for fifteen mins at 2800 rpm in order to extract the serum. Blood samples were stored in a vacutainer serum tube, and the pooled serum that was extracted from them was held there until analysis. The specimen (serum) was then spiked with various concentrations of DENV-Ag.

### 2.3. Synthesis of Gold-Decorated Nanocomposites of Zinc and Graphene Oxide

Production of AuNPs: CTAB and citrate capped-AuNPs are synthesized via a seeded growth approach, as described by Hasan et al. [16] (a) Preparation of seeds—Trisodium citrate (0.0005 M) and HAuCl_4_ (0.0005 M) aqueous solutions were quickly blended for 10 to 15 min. Then, 0.1 M NaBH_4_ (ice-cold solution) was then slowly added after stirring for 20 min, and the mixture was kept undisturbed for two to three hours. (b) Synthesis of growth mixture—two hundred milliliters of 0.0005 mM HAuCl_4_ aqueous solution in a conical flask containing CTAB powder was added, and the combination was heated to fifty–sixty degrees Celsius until the solution turns into a clear orange tone. (c) Growth of seedling—A 0.1 M aqueous ascorbic acid solution was stirred into thirty-six milliliters of the growth solution. Following that, 4 mL of seed solution was gradually added while the mixture was vigorously stirred for ten to twenty minutes.

Synthesis of zinc oxide nanorods: After gold NPs, ZnO-nanorods were synthesized. Zinc nitrate dehydrate (0.004 M) was prepared in double-distilled water and stirred for 2 h, and NaOH (0.007 M) was prepared in double-distilled water for 70 min until it dissolves. After being prepared, NaOH was heated and continuously stirred before being added dropwise to the zinc nitrate dehydrate. To increase the concentration of nanorods, 20 mL of ethanol was also added to the homogenous solution [17].

Synthesis of graphene oxide nanosheets: The process used to synthesize graphene oxide nanosheets includes 5 g of natural graphite powder and 100 mL of cooled, concentrated H_2_SO_4_ that was then combined, and the mixture was homogenized. Potassium permanganate in the amount of 12 g was added gradually with ongoing agitation. The suspension was then agitated for the next three hours at room temperature. The reaction was then given 150 mL of DI water that was stirred for one and a half hours. After that, 75 mL of 30% hydrogen peroxide was added to the mixture. To preserve pH, the product was filtered, immediately washed with hydrochloric acid, and then rinsed with deionized water. To form a brown dispersion, the solution was then diluted with double-distilled water. One further round of filtration was performed on the product before being dried at 80 degrees Celsius overnight to produce powdered graphene oxide nanosheets.

Synthesis of nanocomposites (Au-NP/ZnO-NR/GO-NS): Mix the graphene oxide nanosheets and zinc nanorods first, and then deposit the gold nanoparticles over the surface of the nanocomposites. The ratio of ZnO-NR to GO-NS is 60:40.

### 2.4. Fabrication and Construction of Origami-Platform

A solid skin with a laser-cut design was attached to a silkscreen, which was used for hand printing. The silk screen’s dimensions were predetermined for a 3-E arrangement. Using a squeezer, carbon conductive ink was forced through predetermined openings in an overhead screen and onto cellulose origami sheets. For the preparation of the electrode, a silkscreen was employed as a stencil, and the electrode’s dimensions were then fixed and framed on it. The printed electrodes consisted of three electrodes: a counter electrode (CE), a working electrode (WE), and a reference electrode (RE) drop cast with Ag/AgCl. Cut-outs (channels) were made on the blank section using conventional punching techniques. A piece of paper substrate comprising of a reference microchannel, working sample loading pad, and patterning electrodes was sequentially folded to construct the origami device, and a potentiostat was connected to the sealed OBA to start the sensing process. The blank section was folded downwards, and the inlets were aligned with WE, CE, and RE. A large section for WE was designed onto the blank area of the OBA to employ nanomaterials and aptamer efficiently for DENV diagnosis. This results in the manufacture of the origami-based biosensor. Figure 1 depicts the fabrication and construction of the origami platform.

### 2.5. Procedure for Deposition of Gold-Decorated Zinc and Graphene Oxide Nanocomposite and Immobilization on the Origami-Based Aptasensor

Zinc and graphene oxide nanocomposites that had been produced and coated with gold were deposited (30 µL) onto the WE. The paper electrode was then allowed to dry overnight. After being modified using Au/ZnO/GO-NC, the DENV aptamer (20 µL) was surface immobilized onto the substrate via physio-adsorption. Further applications of these nanocomposite/aptamer-modified electrodes included the detection of the DENV antigen.

### 2.6. Binding of the DENV-Ag and DENV-Apt/Au/ZnO/GO-NC/oPAD

DENV-Ag (0.1 mg/mL to 1000 mg/mL) were evaluated at various concentrations. To establish the interaction between the DENV antigen and the DENV aptamer, electrochemical measurements were conducted.

### 2.7. Stages for Electrochemical Detection

With the use of nanocomposites, the aptamer was deposited onto the surface of the WE to develop a functional and selective biosensor for the detection of the dengue virus. The CV values of electrodes with no NC deposits (bare electrodes) were evaluated for this. The Au/ZnO/GO-NC was then applied to the paper-based aptasensor and allowed to dry for the next day. After that, voltammetry analysis was performed. The subsequent stage involved depositing the aptamer onto the OBA consisting of dried Au/ZnO/GO-NC while simultaneously recording the CV readings.

### 2.8. Optimization of Physio-Chemical Parameters to Analyse Origami-Based Aptasensor

By observing the variations in the voltammograms caused by altering the various parameters, the origami-based aptasensor’s detecting capabilities were modified. The dengue antigen was produced at concentrations of 0.0001, 0.001, 0.01, 0.1, and 1 mg/mL, incubated at a range of temperatures (10, 20, 30, and 40 °C), and the potentiostat response time (*tcond*) was adjusted.

### 2.9. Process for Human Serum Analysis, Repeatability, Reproducibility, and Stability Analysis for DENV-Ag-Apt/Au/ZnO/GO-NC/oPAD

The capability of the aptasensor to work in real-world samples was evaluated by spiking a set concentration of antigen into a healthy human serum. This solution, together with the hybridization indication, was applied to OBA. To substantiate the results, electrochemical assessments were carried out. The reproducibility of the sensor was checked repeatedly (i.e., *n* = 5), and measurements were performed at various times under identical parameters every week to establish the precise target value, and its stability was confirmed for at least a month.

### 2.10. Principle behind Sensing

An ionic indicator (MB) was used for the detection of an aptamer–antigen binding. When Methylene Blue intercalates with the aptamer’s free G nucleotide bases, the electrochemical process is accelerated, and the current response amplifies. A change in the ability to transmit electrons between an electrode surface and a redox molecule occurs when the flexibility and/or conformation of the aptamer is altered following target hybridization (Figure 2) [18]. As a result, steric hindrance is induced by the antigen acting as an insulator and the current response is decreased.

## 3. Result and Discussion

Firstly, Au/ZnO/GO-nanocomposite is deposited onto the surface of the bare working electrode. The aptamer was also immobilized on the working electrode following the drop-deposition of Au/ZnO/GO-NC. The Au/ZnO/GO-NC contributes to the transmission of electrons while also creating a biocompatible environment for the bioreceptor.

### 3.1. Characterization of Au/ZnO/GO-Nanocomposites

The effective preparation of Au/ZnO/GO-nanocomposites was validated via the subsequent characterization strategies: FESEM, FTIR, and UV-Vis spectroscopy, the most important tool for confirming the nanostructure’s synthesis. The morphological confirmation of the Au-modified ZnO-GO NC was investigated by FESEM micrographs. Figure 3a showed the uniformly distributed bright spheres of AuNPs with an average diameter of 30–50 nm. Figure 3b,c represent the successful formation of graphene oxide nanosheets and ZnO nanorods. The obtained ZnO nanorods have an average length of 500 nm with a diameter of 50 nm. Figure 3d showed the presence of Au nanoparticles distributed evenly between ZnO nanorods and graphene oxide nanosheets, respectively.

The FTIR spectrum of the Au-modified ZnO-graphene oxide nanocomposite is displayed in Figure 4a. The various moieties of oxygen are represented by the peaks at (C-O-C) (1210–1320 cm^−1^), (COOH) (1650–1750 cm^−1^), and (C=O) (1450–1650 cm^−1^). The reduction of Au^3+^ to Au^0^ results in strong peaks at 2925 cm^−1^ and 2851 cm^−1^ due to the C-H stretching of alkane and aldehyde. The peaks at 954 and 1039 cm^−1^ can be assigned to the C-H stretching and C=C aromatic stretching. All the hydroxyl group vibrations represented peaks between 3000–3800 cm^−1^ [19,20]. The UV-Vis absorption spectrum of Au-modified ZnO-graphene oxide NC is shown in Figure 4b. The peak at 275 nm was assigned to the π–π transition of C-C and C=C bonds in sp^2^ Carbon. The characteristic peak of ZnO appeared at 370 nm, confirming the presence of ZnO in the nanocomposite. Furthermore, a small plasmonic peak appeared at 534 nm, which represents the presence of gold nanoparticles in the Au-modified ZnO-graphene oxide nanocomposite [21,22,23].

### 3.2. Electrochemical Properties of DENV-Ag-Apt/Au/ZnO/GO-NC/oPAD

The electrochemical cyclic voltammetry approach was used to characterize the electrochemical properties of DENV-Ag-Apt/Au/ZnO/GO-NC/OBA-modified electrodes. Figure 5a depicts the CV-validated differential current response at various phases of the electrode. The CV-bare electrode (oPAD) had a lower current response, which can be ascribed to slower electron transfer kinetics. The rapid electron transfer kinetics given by Au/ZnO/GO-NC resulted in the more-than-sixfold amplification of the current responsiveness upon the deposition onto the working surface. Due to the non-conducive nature of the aptamer, the current is intensely decreased after the immobilization of the bio-receptor (Aptamer) onto the working surface. The current response was further reduced with the addition of antigen. The hybridized antigen–aptamer substantially lowered the current response. Figure 5b shows the bar graph of CV analysis. (Table 1 Depicts the current range). In Figure 5c, the electrochemical impedance spectroscopy, i.e., EIS, was also employed to investigate the surface behavior of the modified electrode of origami-based aptasensors for the detection of DENV. The semi-circle arc represented electron transmission in a restricted way at higher frequencies, whilst the linear component of the graph represented diffusion behavior at lower frequencies.

The charge transfer resistance (Rct) value is provided by the Nyquist diameter, which is measured by the real axis value at the lower frequency intercept. Rct is the interference provided by the electrode material in transferring electrons from the solution mixture to the electrode, as compared to the surface modification. The typical curve shows that the Rct of the bare electrode (oPAD) is substantially greater than the Rct of the ternary nanostructure (Au/ZnO/GO-NC/oPAD)-coated electrode due to the improved electron transfer kinetics. Because of the non-conductive behavior of the aptamer (Apt/Au/ZnO/GO-NC/oPAD), the Rct value increased after aptamer immobilization on the electrodes of the ternary nanostructure-coated surface when compared to the ternary nanostructure coated electrodes. When the functionalized electrode was exposed to the DENV antigen (DENV-Ag-Apt/Au/ZnO/GO-NC/OBAs), the Rct value signal enhanced much more because the DENV antigen bonded to the aptamer and formed a twofold layer of insulator. All EI-spectra were recorded in 10 mM MB utilizing 0.1 M KCl, demonstrating a quick reaction.

### 3.3. Effect of the Various Antigens of DENV Concentrations on the Apt/Au/ZnO/GO-NC/oPAD

To demonstrate the quantitative capability of the proposed aptasensor, various antigen concentrations were examined. Since research is conducted globally in the concentration range of 0.0001 mg/mL to 1 mg/mL, a broad concentration range from this range was employed for aptamer binding. The findings demonstrate the antigen hybridization with the aptamer and varying current responses at various concentrations, validating the proposed sensor’s quantitative performance. The acquired results were consistent with the previously reported sensors. Increased antigen concentrations resulted in a lower current response because electron transfer was hindered by the increased biological recognition element insulating layers. It was discovered that the detection limit was 10^2^ ng/mL. Electrochemical measurements were used to confirm all these prepared concentration values (Figure 6).

### 3.4. Optimization of DENV-Ag-Apt/Au/ZnO/GO-NC/oPAD in Terms of Temperature and Time

For the desired sensor to operate as intended, the biosensor must be optimized. Time and temperature have an impact on the sensor’s functionality. For achieving optimal responsiveness, certain parameters were optimized for the sensor. The performance of the constructed DENV-Ag-Apt/Au/ZnO/GO-NC/oPAD was thoroughly investigated. At varied temperatures ranging from 10 °C to 40 °C, cyclic voltammograms of dengue antigen/aptamer/Au/ZnO/GO-NC/OBAs were detected at 50 mV/s. The highest current amplification was measured at 40 °C; however, it was comparable to the aptamer’s current response. As a result, the aptasensor was calibrated at 30 °C (Figure 7a). The developed sensor was enhanced over a range of time (seconds) to produce the best response feasible at any specific moment. It was discovered that the current rises from 20 to 40 s (Figure 7b), yet 20 s may be considered the best response time since, at 30 and 40 s, the response of the current takes longer to show, delaying the developed origami aptasensors’ capacity to detect changes. As a result, we will be unable to determine it in a shorter period of time.

### 3.5. Detection Limit and Precision/Accuracy (Recovery) Test of the Origami-Based Aptasensor

The lowest antigen concentration that might be detected (0.1 mg/mL) is known as the limit of detection, or LOD. Precision and accuracy were assessed using a recovery test. As a result, various concentrations of antigens have been spiked into the other concentration. To illustrate the biosensors extraordinary recovery, 0.001 mg/mL of the dengue antigen concentration was added to 0.0001 mg/mL of concentration. This reveals that the current is roughly comparable to the concentration of 0.1 mg/mL. Figure 8 depicts the cyclic voltammetry graph of recovery.

### 3.6. Examination of Specificity/Reliability (Cross-Reactivity) and Stability

The performance of the origami-based aptasensors in terms of cross-reactivity was tested using CHIKV 0.1 mg/mL. Figure 9a displays the current responses of the cyclic voltammetry-recorded detection samples. The CHIKV peak flow is relatively similar to that of the Apt/Au/ZnO/GO-NC/oPAD. Additionally, the origami-based aptasensor was stored at 4 °C for several weeks in order to identify the dengue antigen (0.1 mg/mL) and test the stability of the origami-based aptasensor via CV. The results indicated that the constructed sensor remained stable through to day 15 and generated results that were almost equivalent to Apt/Au/ZnO/GO-NC/oPAD. After the 15th day, it starts to decline, as observed in (Figure 9b). The stability graph also displayed the error bar, indicating sufficient reproducibility or repetition for DENV antigen determination.

### 3.7. Investigation in Human Serum (Healthy)

To authenticate the accessibility of the constructed origami-based aptasensor, the possibility that the proposed platform for detecting dengue antigens in human serum samples was evaluated using a standard addition procedure. Peak current cyclic voltammetry value testing was performed after spiking 0.1 mg/mL dengue antigen in the human serum to the surface of Apt/Au/ZnO/GO-NC/oPAD (Figure 10a,b). The sensor performed well in samples of human serum, and the reaction was essentially similar to that of the DENV polyvalent antigen. Data from the actual sample that fulfilled the requirements for identification were obtained, demonstrating the sensor’s efficiency.

### 3.8. Comparative Analysis

Numerous electrochemical biosensors have been developed for the detection of DENV, but the majority of these stated biosensors were based on a bulky sensing tool. A less selective bio-recognition element, i.e., antibodies, also employed a more commercialized electrode setup, which makes these sensors more expensive, while also being less selective and sensitive; they are also not truly appropriate for point-of-care-based approaches. Therefore, in this work, we have self-fabricated a low-cost origami electrode set-up and used highly selective aptamers against DENV, and, to make it more sensitive, we have used ternary nanocomposites of gold-decorated zinc and graphene nanostructures. To the best of our knowledge, this is the first report of dengue electrochemical aptasensors based on self-fabricated origami electrode setups coated with gold-decorated zinc and graphene-based ternary nanostructures. A comparison of the stated electrochemical biosensors for the detection of DENV with the present study were shown in Table 2.

## 4. Conclusions and Future Perspective

Dengue fever has been around for a while, but it is still a major concern worldwide. The early diagnosis and treatment of dengue patients has been a major priority for researchers. The management of dengue can greatly benefit from early and prompt detection. There are numerous reports on commercialized SPEs as a biosensor for the detection of DENV, but these electrodes have a number of drawbacks, including the electrode setup being fully uncovered, the solution being susceptible to outside influences, and the tracks and design of the electrodes not being confidential. Therefore, to overcome these limitations, low-cost origami-based aptasensors were constructed. The use of origami paper electrodes has advantages such as providing a closed chamber to the electrodes, keeping them confidential, and protecting them from dust particles or external factors. As a result, the current study describes the development of OBAs (origami-based aptasensors) for the quick, low-cost, and early detection of dengue. To improve signal responsiveness, Au/ZnO/GO-NC was layered on a three-electrode system. On a single platform, the suggested method can identify all four serotypes of dengue antigens. In future, such low-cost origami designs can be improved by using different patterns of folding techniques, such as the kirigami approach.

## Figures and Tables

**Figure 1 sensors-24-00801-f001:**
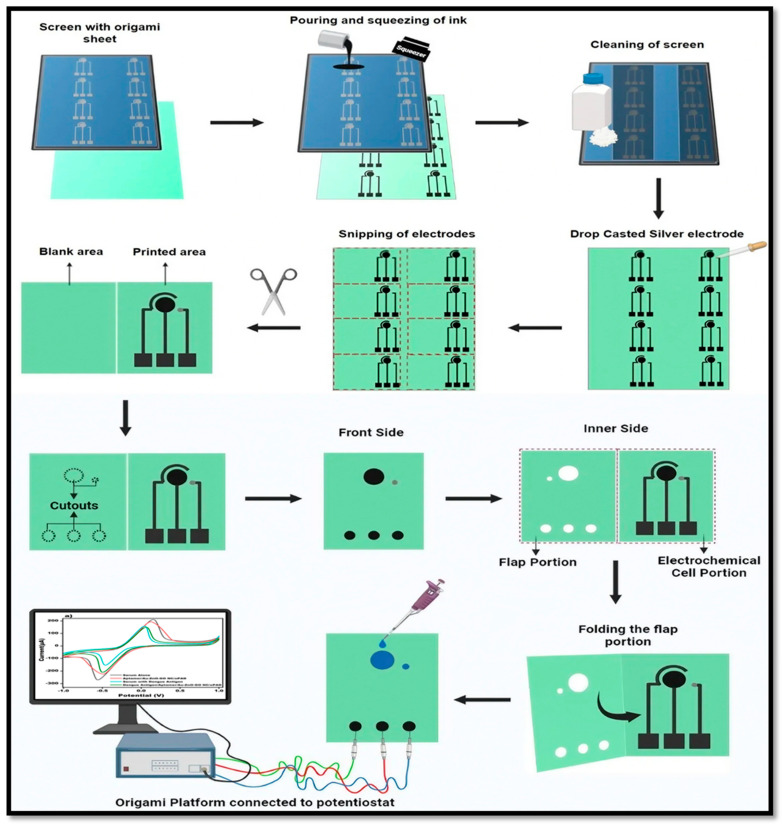
Diagrammatic stepwise representation of fabrication and construction of origami platform: starting with the placing of the origami sheet (used as a substrate) under the screen; the pouring and squeezing of carbon conductive ink used as a base on the screen for the hand printing of electrodes; the cleaning of the screen using acetone and cotton to prevent it from tarnishing; the drop-casting of the silver electrode as a reference electrode; snipping out all the electrodes and constructing cut-outs on the blank section of the flap; connecting the origami platform to the potentiostat for the electrochemical detection of DENV-Ag.

**Figure 2 sensors-24-00801-f002:**
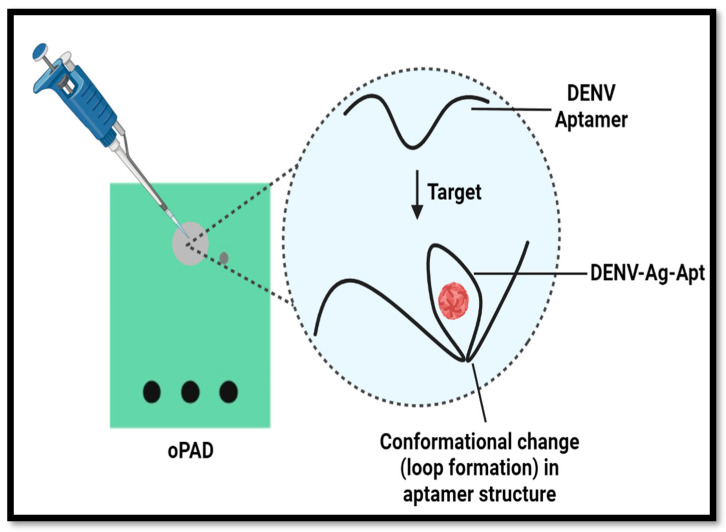
Diagrammatic representation of loop formation by aptamer.

**Figure 3 sensors-24-00801-f003:**
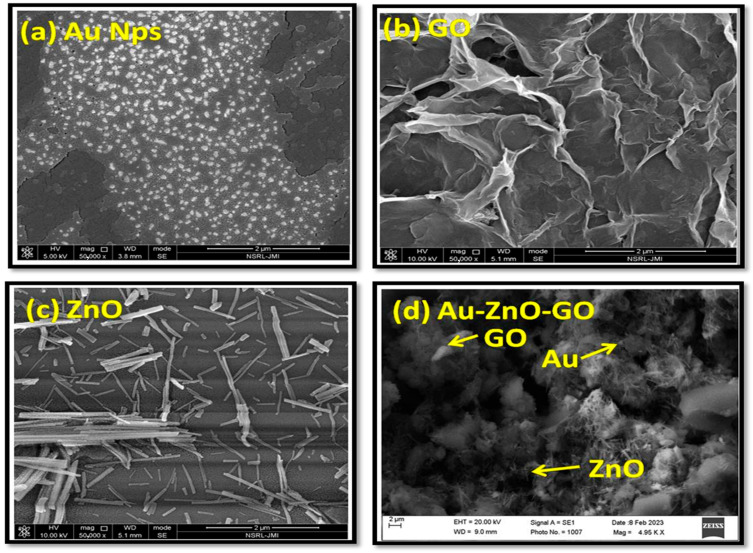
Field emission scanning electron microscopy (FESEM) micrographs of (**a**) bright spheres of gold nanoparticles (AuNPs) ranging an average of 30–50 nm (**b**) graphene oxide nanosheets (GO Ns) (**c**) ZnO nanorods (ZnO nR) with an average diameter of 50 nm and an average length of 500 nm, and (**d**) the successful formation Au-ZnO-GO nanocomposite.

**Figure 4 sensors-24-00801-f004:**
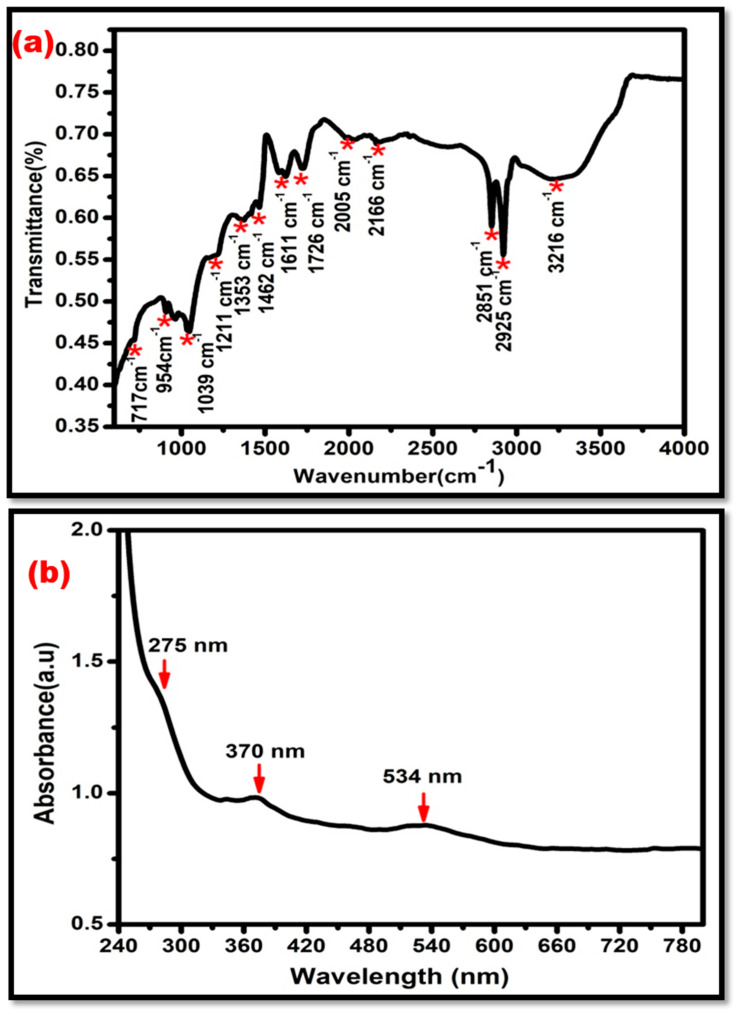
(**a**) Fourier transform infrared (FTIR spectrum) spectroscopy of AU-ZnO-GO NC with strong peaks at 2925 cm^−1^ and 2851 cm^−1^, due to the reduction of Au^3+^ to Au^0^. (**b**) UV-Vis absorption spectrum of Au-modified ZnO-GO NC, showing characteristic peaks at 275 nm, 370 nm, and 534 nm.

**Figure 5 sensors-24-00801-f005:**
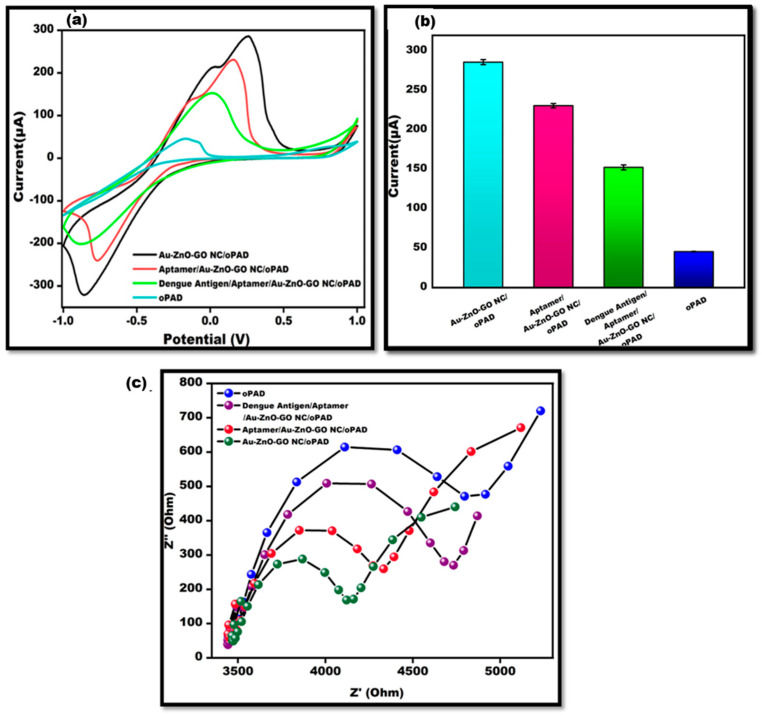
(**a**). Cyclic voltammetry graph of bare oPAD, Au/ZnO/GO-NC/oPAD, Apt/Au/ZnO/GO-NC/oPAD, and DENV-Ag-Apt/Au/ZnO/GO-NC/oPAD utilizing 10 mM MB in 0.1 M KCl at 50 mVs^−1^ in the potential range from −1 V to +1 V. (**b**). Bar graph representing oPAD, Au/ZnO/GO-NC/oPAD, Apt/Au/ZnO/GO-NC/oPAD, and DENV-Ag-Apt/Au/ZnO/GO-NC/oPAD with an error bar *n* = 5. (**c**). Comparative EIS spectra depicting oPAD, Au/ZnO/GO-NC/oPAD, Apt/Au/ZnO/GO-NC/oPAD, and DENV-Ag-Apt/Au/ZnO/GO-NC.

**Figure 6 sensors-24-00801-f006:**
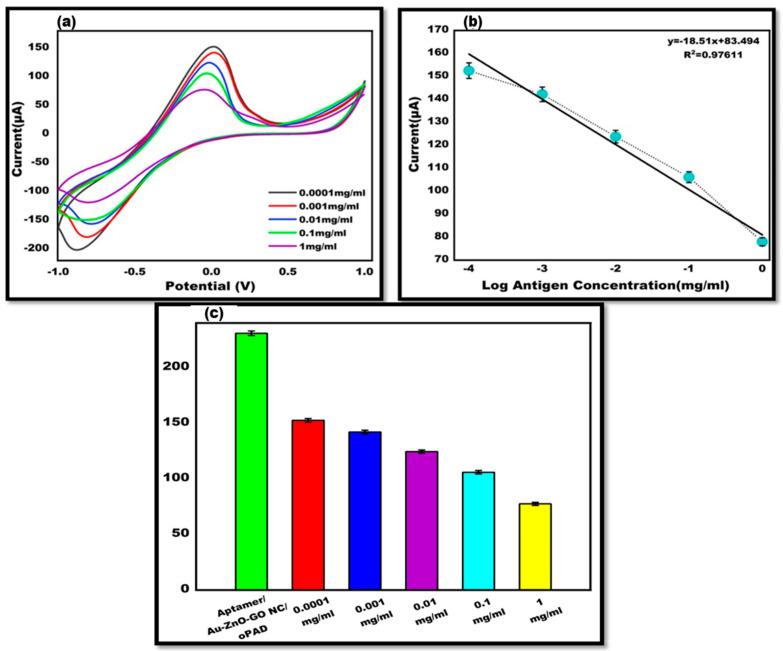
(**a**) Cyclic voltammetry of DENV-Ag-Apt/Au/ZnO/GO-NC/oPAD-adapted electrode, utilizing various antigen concentrations such as 0.0001 mg/mL^−1^ mg/mL in 10 mM MB in 0.1 M KCl at 50 mVs^−1^ in the potential range from –1 V to +1 V. (**b**) Linearity of cyclic voltammetry with log antigen concentration. (**c**) Bar graph of various Ag concentrations ranging from 0.0001–1 mg/mL along with DENV-Apt/Au/ZnO/GO-NC/oPAD.

**Figure 7 sensors-24-00801-f007:**
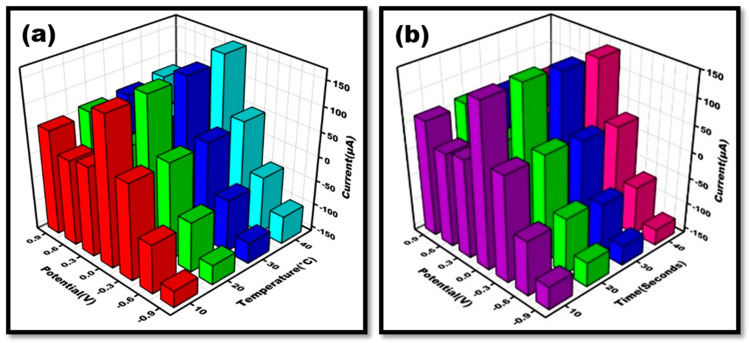
Three-dimensional illustration of cyclic voltammetry achieved at DENV-Ag-Apt/Au/ZnO/GO-NC/oPAD for (**a**). Different temperatures ranging from 10–40 °C in 10 mM Methylene blue dissolved in 0.1 M KCl at 50 mVs^−1^. *(***b**). Time (tcond ranging from 10–40 s) in 10 mM Methylene blue dissolved in 0.1 M KCl at scan rate 50 mVs^−1^.

**Figure 8 sensors-24-00801-f008:**
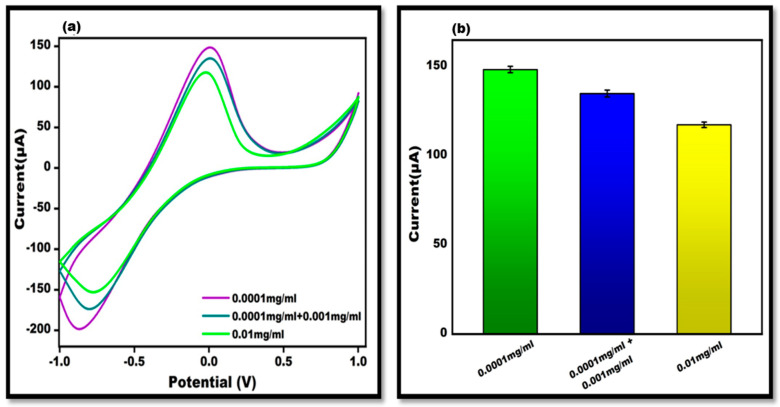
(**a**). Cyclic voltammetry graph illustrating the recovery value of DENV-Ag-Apt/Au/ZnO/GO-NC/oPAD in 10 mM Methylene blue in 0.1 M KCl at 50 mVs^−1^ in the potential range from −1 V to +1 V. (**b**). Bar graph representing the recovery values of DENV-Ag-Apt/Au/ZnO/GO-NC/oPAD in 10 mM Methylene blue in 0.1 M KCl at 50 mVs^−1^ in the potential range from −1 V to +1 V with an error bar, *n* = 5.

**Figure 9 sensors-24-00801-f009:**
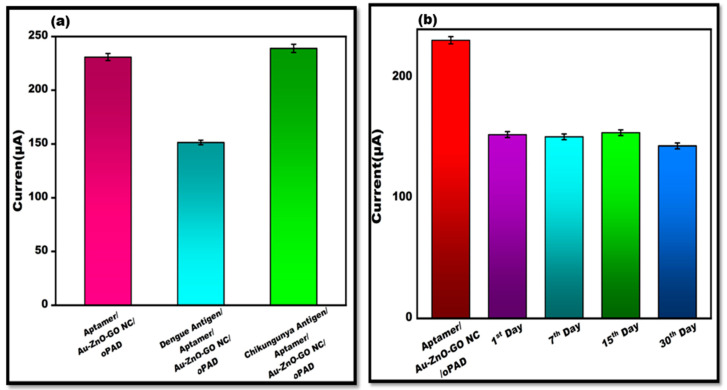
(**a**). Evaluation of peak current CV value of DENV-Ag-Apt/Au/ZnO/GO-NC/oPAD binding with the interferent, CHIKV-Ag with an error bar, *n* = 5. (**b**). The stability of the developed OBA was evaluated using the electrochemical test for dengue antigen on the 4 weeks with error bar (*n* = 5).

**Figure 10 sensors-24-00801-f010:**
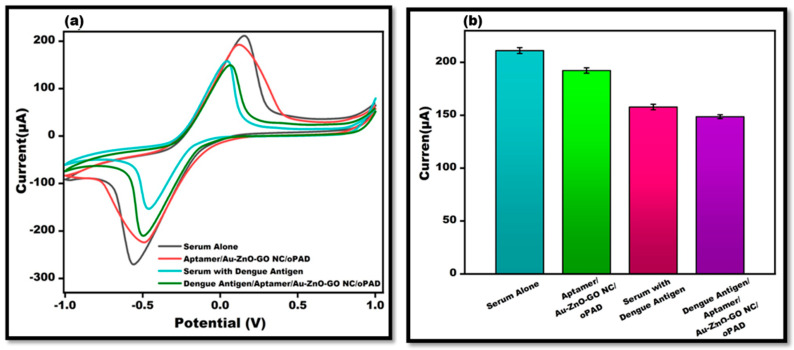
(**a**). CV peak current evaluation of DENV-Ag in the sample using the designed OBA depicting serum alone, Apt/Au/ZnO/GO-NC/oPAD. Serum with DENV-Ag and DENV-Ag-Apt/Au/ZnO/GO-NC/oPAD. (**b**). Bar graph of DENV-Ag in human serum using the fabricated origami-based aptasensor with an error bar (*n* = 5).

**Table 1 sensors-24-00801-t001:** Representing different stages of current response of bare oPAD, Au/ZnO/GO-NC/oPAD, Aptamer/Au/ZnO/GO-NC/oPAD, and DENV-Ag-Apt/Au/ZnO/GO-NC/oPAD in a tabular form.

S. No.	Different Stages of Origami-Based Aptasensor	Current Response
1.	oPAD	45.11 microamperes
2.	Au/ZnO/GO-NC/oPAD	285.95 microamperes
3.	Aptamer/Au/ZnO/GO-NC/oPAD	230.84 microamperes
4.	DENV-Ag-Apt/Au/ZnO/GO-NC/oPAD (0.0001 mg/mL)	152.52 microamperes

**Table 2 sensors-24-00801-t002:** The efficiency of several electrochemical biosensors for detecting DENV is compared using the LOD, nanomaterial, and electrode platform.

S.No.	Electrochemical Biosensor/Platform	DENV-Biomarker	Nanomaterial	LOD (ng/mL)	References
1.	Multiplex-genosensor based on paper electrode	DENV-all serotypes	GO-SiO_2_–Nanocomposite	44.50	[24]
2.	EIS-based immunosensor	DENV-NS1	Gold nanoparticles and polyaniline-based nanocomposites	1.8 × 10^−6^	[25]
3.	Impedimetric immunosensor	NS 1	Gold nanorod-decorated graphitic carbon nitride	0.09	[26]
4.	Immunosensor	NS 1	Ternary nanocomposite of reduced graphene oxide, polydopamine, and gold nanoparticles	1.78	[27]
5.	Screenprinted electrode-based immunosensor	Dengue virus antigen	Not used	0.0461 in human plasma	[28]
6.	3D-printed paper-based aptasensor	Polyvalent DENV antigen	ZnO-NPs	100	[29]
7.	Origami-based aptasensor	Polyvalent DENV antigen	Gold-decorated zinc and graphene oxide-based bernary nanostructure	100	This work

## Data Availability

Data are contained within the article.

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
