# Peer review of "Ternary Nanostructure Coupling Flip-Flap Origami-Based Aptasensor for the Detection of Dengue Virus Antigens"

_sensors, 2024, doi:10.3390/s24030801_

Round 1

Reviewer 1 Report

Comments and Suggestions for Authors

This work can be accepted in the present form.

Author Response

attached file

Reviewer 2 Report

Comments and Suggestions for Authors

The paper “Ternary Nanostructure Coupling Flip-Flap Origami-Based Aptasensor for the Detection of Dengue Virus Antigen” by Mohd. Rahil Hasan and coauthors describes the development of the paper-based electrochemical aptasensor for the detection of the polyvalent antigen DENV.

A gold/Zn/graphene nanocomposite was used as the carrier of the analytical system; recognition and binding of the DENV antigen occurs due to a highly specific aptamer pre-sorbed here. The goal of the work is to create an effective test system suitable for use in POC conditions. It is important and relevant.

Some questions and comments:

1. Figure 5b: concentration of DENV varies in a range of 4 logarithmic units whereas the signal values – only in range of 2 times… Figure 5b: Ag-concentrations of 0.0001 mg/ml sharply change the signal magnitude, while a further sharp increase in antigen concentration does not cause such a drop. Is it normal? This needs to be discussed somehow.

  1. . Table 1, line 4 (DENV-Ag-Apt/Au/ZnO/GO-NC/oPAD): the target concentration (DENV-Ag) must be indicated.
  2. The term “Real sample” is not entirely correct - you did not use patient sera, but artificial samples prepared from sera and targets.
  3. References list should be compiled according to the Journal list.

Thus, the submitted manuscript needs some revision.

Author Response

attached file

Reviewer 3 Report

Comments and Suggestions for Authors

Please find attached comments and suggestions. 

Comments on the Quality of English Language

 Minor editing of English language required

Author Response

attached file

Round 2

Reviewer 3 Report

Comments and Suggestions for Authors

Please find attached review report

Comments on the Quality of English Language

Grammatical errors should be improved.

Author Response

attached file
